# First-Principles Study of Doped *CdX*(*X* = *Te*, *Se*) Compounds: Enhancing Thermoelectric Properties

**DOI:** 10.3390/ma17081797

**Published:** 2024-04-14

**Authors:** Junfeng Jin, Fang Lv, Wei Cao, Ziyu Wang

**Affiliations:** 1The Institute of Technological Sciences, Wuhan University, Wuhan 430060, China; 2021206520035@whu.edu.cn; 2Key Laboratory of Artificial Micro- and Nano- Structures of Ministry of Education, School of Physics and Technology, Wuhan University, Wuhan 430072, China; fanglv@whu.edu.cn

**Keywords:** thermoelectric, doped *CdX* compounds, first-principles calculations, isovalent doping

## Abstract

Isovalent doping offers a method to enhance the thermoelectric properties of semiconductors, yet its influence on the phonon structure and propagation is often overlooked. Here, we take CdX (X=Te, Se) compounds as an example to study the role of isovalent doping in thermoelectrics by first-principles calculations in combination with the Boltzmann transport theory. The electronic and phononic properties of $Cd_{8}Se_{8}$, $Cd_{8}Se_{7}Te$, $Cd_{8}Te_{8}$, and ${Cd}_{8}{Te}_{7}Se$ are compared. The results suggest that isovalent doping with CdX significantly improves the thermoelectric performance. Due to the similar properties of Se and Te atoms, the electronic properties remain unaffected. Moreover, doping enhances anharmonic phonon scattering, leading to a reduction in lattice thermal conductivity. Our results show that optimized p-type(n-type) ZT values can reach 3.13 (1.33) and 2.51 (1.21) for ${Cd}_{8}{Te}_{7}Se$ and $Cd_{8}Se_{7}Te$ at 900 K, respectively. This research illuminates the potential benefits of strategically employing isovalent doping to enhance the thermoelectric properties of CdX compounds.

## 1. Introduction

One of the primary sbyproducts of using various energy forms is heat. The process of converting this excess heat into electrical energy, known as thermoelectricity, is seen as a promising technology for practical energy harvesting applications [1]. The efficiency of thermoelectric conversion is assessed using the dimensionless thermoelectric figure of merit ZT [2]. ZT is defined as
(1)$$ZT=\frac{S^{2}\sigma T}{\kappa_{e}+\kappa_{l}}$$
where σ, *S*, *T*, $\kappa_{e}$ and $\kappa_{l}$ represent electrical conductivity, Seebeck coefficient, temperature, electronic thermal conductivity, and lattice thermal conductivity, respectively. However, the coupling effects in thermoelectric performance make it challenging to directly enhance the thermoelectric properties [3]. This is due to the intricate interplay between electrical conductivity, thermal conductivity, and Seebeck coefficient, which are often coupled together. Improving one of these properties can inadvertently affect the others, making it difficult to achieve substantial enhancements in overall thermoelectric performance without carefully considering and addressing these interdependent factors. Consequently, targeted strategies that can effectively decouple these properties or optimize their collective interaction are crucial for achieving significant advancements in thermoelectric materials. Various strategies, including doping [4], band engineering [5], phonon engineering [6], nanostructuring [7], and alloying [8], have been proposed to enhance thermoelectric performance.

Doping in thermoelectric materials enables precise tuning of electronic properties, providing flexibility, versatility, and compatibility with other enhancement techniques. There are many types of doping in thermoelectric materials, including cationic doping [9], co-doping [10], ion doping [11] and single-atom doping [12]. Here, we primarily focus on single-atom doping. Based on the doping atoms, doping in thermoelectric materials can be classified into aliovalent doping, which involves introducing impurities of different valences, and isovalent doping, which involves introducing impurities of the same valence. Aliovalent doping is commonly utilized to regulate carrier concentration for optimizing ZT. Research on the effects of Nb doping on the thermoelectric properties of n-type half-Heusler compounds revealed an enhanced power factor and a 20% increase attributed to aliovalent doping-induced decoupling between thermoelectric parameters [13]. Han et al. [14] emphasized the critical impact of aliovalent dopants on controlling the phonon structure and inhibiting the phonon propagation in a heavy-band NbFeSb system. Baranets et al. [15] demonstrated that aliovalent substitutions can alter the dimensionality of the polyanionic sublattice in the resulting quaternary phases, leading to reduced electrical resistivity and a notably enhanced Seebeck coefficient.

Compared to aliovalent doping, isovalent doping ideally decouples and regulates thermoelectric performance by reducing thermal conductivity through phonon scattering while maintaining unchanged electronic properties. Musah et al. [16] summarized a review of isovalent substitution as a method to independently enhance thermoelectric performance and device applications. The substitution of isovalent ions in the anion Te-site of Bi–Sb–Te led to a significant enhancement of the ZT over a wide temperature range, with the ZT being increased by 10% for all measured temperatures and averaging beyond 1.0 between 300 and 520 K, demonstrating the synergetic control of band structure and deformation potential via isovalent substitution [17]. He et al. [18] also demonstrated that isovalent Te substitution effectively reduces $\kappa_{l}$ and increases σ in hole carrier concentration.

To thoroughly explore the impact of equiatomic doping on regulating thermoelectric performance, we chose CdX (X=Se,Te) as the focus of our research and utilized a first-principles approach. Recent research [19,20] indicated that CdX is commonly used as a dopant in thermoelectric applications. The simple cubic phase structure of CdX provides advantages for first-principles studies due to its well-defined symmetry and straightforward electronic and phononic property calculations. Additionally, Te and Se share similarities in their doping characteristics, owing to their comparable chemical properties and the analogous effects they induce when integrated into host materials. In this paper, we systematically investigated the electronic, phononic, mechanical, bonding, and thermoelectric properties of CdX using first-principles combined with Boltzmann transport theory.

## 2. Computational Methods

Theoretical computations were conducted using density functional theory (DFT) within the Quantum ESPRESSO v6.2 (QE) code [21,22]. The exchange–correlation functional used is the Generalized Gradient Approximation (GGA) as given by Perdew–Burke–Ernzerhof (PBE) [23], and the corresponding pseudopotential files are sourced from the standard solid-state pseudopotentials (SSSP PBE Efficency v1.3.0) library [24]. A kinetic energy cut-off of 80 Ry was utilized, and all relaxations were carried out until the forces and energy on each atom were reduced to less than 10^−4^ Ry/Bohr and 10^−10^ Ry. The Heyd–Scuseria–Ernzerhof (HSE06) hybrid functional [25,26] was used to obtain a more accurate band structure for the primitive cell of CdX. We constructed the doping structure using a 2 × 2 × 2 supercell of the primitive cell for both CdTe and CdSe. The Brillouin zone was sampled over a uniform Γ-centered k-mesh of 4 × 4 × 4. The projected crystal orbital Hamilton population (COHP) was calculated using the LOBSTER [27,28] package. The mechanical properties were carried out using Voigt–Reuss–Hill approximation [29], as implemented in the ElATools v1.7.0 [30] package. The crystal structure was plotted using VESTA v3.5.7 software [31].

Boltzmann’s transport theory was employed to analyze the transport properties of systems using the BoltzTraP code [32]. Under Boltzmann’s transport theory, these electronic transport coefficients can be expressed as
(2)$$S_{\alpha \beta }\left(T,\mu \right)=\frac{1}{eT}\frac{\int \nu_{\alpha }\left(i,k\right)\nu_{\beta }\left(i,k\right)\left(\varepsilon -\mu \right)\left[-\frac{\partial f_{\mu }\left(T,\varepsilon \right)}{\partial \varepsilon }\right]d\varepsilon }{\int \nu_{\alpha }\left(i,k\right)\nu_{\beta }\left(i,k\right)\left[-\frac{\partial f_{\mu }\left(T,\varepsilon \right)}{\partial \varepsilon }\right]d\varepsilon }$$
(3)$$\frac{\sigma_{\alpha \beta }\left(T,\mu \right)}{\tau_{e}\left(i,k\right)}=\frac{1}{V}\int e^{2}\nu_{\alpha }\left(i,k\right)\nu_{\beta }\left(i,k\right)\left[-\frac{\partial f_{\mu }\left(T,\varepsilon \right)}{\partial \varepsilon }\right]d\varepsilon$$
(4)$$\frac{\kappa_{\alpha \beta }^{e}\left(T,\mu \right)}{\tau (i,k)}=\frac{1}{TV}\int \nu_{\alpha }\left(i,k\right)\nu_{\beta }\left(i,k\right){\left(\varepsilon -\mu \right)}^{2}\left[-\frac{\partial f_{\mu }\left(T,\varepsilon \right)}{\partial \varepsilon }\right]d\varepsilon$$
where α, β are Cartesian components, μ is the chemical potential of electrons (the Fermi level), *V* is volume of the unit cell, *e* is electronic charge, ε is the band eigenvalue, $\nu_{\alpha }\left(i,k\right)$ is the electron group velocity, and $f_{\mu }\left(T,\varepsilon \right)$ is is the Fermi–Dirac distribution.

The carrier relaxation time ($\tau_{e}$) under the electron–phonon averaged (EPA) approximation was evaluated using the following equation [33]:(5)$$\begin{array}{cc} \tau_{e}^{-1}\left(\varepsilon ,\mu ,T\right)=\frac{2\pi \Omega }{g_{s\hbar }}\sum_{\nu } & \left\{g_{\nu }^{2}\left(\varepsilon ,\varepsilon +\bar{\omega }\nu \left[n\left(\bar{\omega }\nu ,T\right)+f\left(\varepsilon +\bar{\omega }\nu ,\mu ,T\right)\right]\times \rho \left(\varepsilon +\bar{\omega }\nu \right)\right.\right.\\ \; & +g_{\nu }^{2}\left(\varepsilon ,\varepsilon -\bar{\omega }\nu \right)\left[n\left(\bar{\omega }\nu ,T\right)+1-f\left(\varepsilon -\bar{\omega }\nu ,\mu ,T\right)\right]\rho \left(\varepsilon -\bar{\omega }*\nu \right)\} \end{array}$$
Here, ε is the energy of the carriers, μ is the chemical potential, Ω is the volume of the primitive unit cell, *ℏ* is the reduced Planck’s constant, $g_{s}$ is the spin degeneracy, ν is the phonon mode index, $g_{\nu }^{2}$ is the averaged electron–phonon matrix, $\bar{\omega }\nu$ is the averaged phonon mode energy, $n\left(\bar{\omega }\nu ,T\right)$ is the Bose–Einstein distribution function, $f\left(\varepsilon +{\bar{\omega }}_{\nu },\mu ,T\right)$ is the Fermi–Dirac distribution function, and ρ is the density of states per unit energy and unit volume.

The lattice thermal conductivity, $\kappa_{l}$, is computed using the Boltzmann transport equation integrated within the ShengBTE code [34], incorporating second- and third-order interatomic force constants (IFCs). The lattice thermal conductivity component $\kappa_{l}^{\alpha \beta }$ (α,β represents three Cartesian axes) is given by
(6)$$\kappa_{l}^{\alpha \beta }=\frac{1}{k_{B}T^{2}\Omega N}\sum_{\lambda }f_{0}\left(f_{0}+1\right){\left(\hbar \omega_{\lambda }\right)}^{2}\upsilon_{\lambda }^{\alpha }\upsilon_{\lambda }^{\beta }\tau_{\lambda }^{0}$$
where Ω, *N*, $f_{0}$, $\omega_{\lambda }$, $\upsilon_{\lambda }$, and $\tau_{\lambda }^{0}$ are volume, number of phonon vectors, Bose–Einstein distribution function, frequency, group velocity, and lifetime of phonon mode λ, respectively. The second-order and third-order IFCS were calculated by a 2 × 2 × 2 supercell, including 128 atoms. The third-order IFCS took the 5th nearest neighbor into consideration. The grid mesh for the phonon was set to 20 × 20 × 20 to obtain convergent lattice thermal conductivity.

## 3. Results and Discussion

### 3.1. Electronic Properties

CdX adopts a zincblende, sphalerite structure and crystallizes in the cubic $F4\bar{3}m$ space group, as depicted in Figure 1a,d. Each $X^{2-}$ ion is bonded to four equivalent $Cd^{2+}$ atoms to form corner-sharing $XCd_{4}$ tetrahedra. To facilitate our study, we constructed a 2 × 2 × 2 supercell of CdX, denoted as $Cd_{8}X_{8}$ as shown in Figure 1b,e, and replaced one *X* atom. The resulting doped structures are illustrated in Figure 1c,f. The relaxed latice constants are also given in Figure 1. The lattice constant and bond length of $Cd_{8}Te_{8}$ are longer than those of $Cd_{8}Se_{8}$, indicating a stronger bond strength in Cd–Se. Upon doping, the lattice constant of $Cd_{8}Se_{7}Te$ increases, while that of $Cd_{8}Te_{7}Se$ decreases.

The band structures depicted in Figure 2a–f all display a similar band shape, with the only distinguishing factor being the band gap. It is evident that the band structures obtained from the HSE06 method exhibit a similar shape to those obtained from the PBE method, with notable differences observed in the band gaps. Specifically, the band gaps for CdSe and CdTe computed using the PBE method are reported as 0.47 eV and 0.58 eV, respectively, whereas those computed using the HSE06 method are reported as 1.42 eV and 1.34 eV, respectively. These findings closely align with the results reported in Ref. [35]. Following doping, there is a reduction in the band gaps. In the case of $Cd_{8}Se_{7}Te$, the band gap reduced to 0.372 eV, and for $Cd_{8}Te_{7}Se$, it reduced to 0.441 eV. Notably, both the valence band maximum (VBM) and conduction band minimum (CBM) are situated at the Γ point. Furthermore, the valence band demonstrates multiple valleys. Similar trends in band structure changes have been observed in other isovalent doped systems [36]. The relationship between band gaps and composition in these systems can be characterized by the quadratic Vegard’s law as [36]
(7)$$E_{g}^{A_{1-x}B_{x}}\left(x\right)=\left(1-x\right)E_{g}^{A}+xE_{g}^{B}-bx\left(1-x\right)$$
where $E_{g}^{A}$ and $E_{g}^{B}$ are the band gaps of the host materials, *A* and *B*, respectively, *x* is the composition, and *b* is a bowing parameter. In our case, *A* and *B* represent CdSe and CdTe, respectively, with *x* equal to 1/8. By fitting bowing parameter *b*, we found it to be 1.037 eV for $Cd_{8}Se_{7}Te$. *b* through $Cd_{8}Se_{7}Te$ is 1.037 eV. Subsequently, we applied this model and fitted *b* to $Cd_{8}Te_{7}Se$ and obtained a band gap of 0.453 eV, which closely aligns with the calculated value. This suggests that the band gap of doped CdX can be predicted using Vegard’s law.

To gain a comprehensive understanding of the band structure, we present the projected band structures of $Cd_{8}Se_{7}Te$ and $Cd_{8}Te_{7}Se$ in Figure 3. The dot size in the projected band structure represents the contribution of corresponding orbitals. The conduction band is primarily composed of the 5 *s* orbitals of Cd. For CdX or doped CdX, their conduction band is the same, while the valence band is dominated by the *p* and *d* orbitals of all atoms. Notably, both Se and Te atoms demonstrate similar contributions, with the *s* orbitals of Se being a little stronger than those of Te, and the *p* orbitals of Se being a little weaker than those of Te. Although isovalent atoms contribute to the band structure, their effect is relatively subtle.

Figure 4 illustrates the calculated values of *S*, σ, and $S^{2}\sigma$ at 300 K for different carrier concentrations. Generally, *S* can be expressed as [37]
(8)$$S=\frac{8\pi^{2}k_{B}^{2}}{3eh^{2}}m^{*}T{\left(\frac{\pi }{3n}\right)}^{2/3}$$
in which $k_{B}$, *e*, $m^{*}$, and *h* are the Boltzmann constant, electron charge, effective mass, and Planck constant, respectively. Analysis of Figure 4a,d reveals that the absolute values of *S* all decrease as the carrier concentration increases. Due to their similar band curvatures ($m^{*}$), CdX exhibits comparable *S* values. Notably, the *S* for hole doping (p-type) is significantly higher than that for electron doping (n-type). For instance, p-type *S* can reach 400 μV/K, while n-type *S* is only 100 μV/K at 10^19^ cm^−3^. This difference can be attributed to the valence band having a much sharper curvature and a larger $m^{*}$ compared to the conduction band. The behavior of σ as a function of carrier concentration is depicted in Figure 4b,e. In contrast to *S*, all σ values increase as the carrier concentration rises. N-type σ is higher than p-type σ, especially at low carrier concentrations. When the carrier concentration reaches to 10^20^–10^21^ cm^−3^, both n-type and p-type σ reach the 10^5^ S/m level. Due to the Wiedemann–Franz relation [38], $\kappa_{e}$ exhibits a linear correlation with σ; hence, there is no need to separately display $\kappa_{e}$. In Ref. [39], the electrical parameters of various thin film CdSe samples were investigated. The carrier concentration of thin film CdSe was found to be approximately (10^20^ cm^−3^) with a Seebeck coefficient of around (−50 μV/K). Our obtained value of (−25 μV/K) aligns closely with this result. It is worth noting that our calculated electrical conductivity (10^5^ S/m) significantly exceeds the experimental value (10^2^ S/m). This discrepancy can be attributed to the DFT simulation assuming a perfect crystal structure, while experimental samples typically exhibit boundaries, grain effects, and scattering mechanisms that substantially reduce the electrical conductivity.

When considering *S* and σ, it is evident that the n-type $S^{2}\sigma$ is significantly lower than p-type $S^{2}\sigma$, as depicted in Figure 4e,f. The optimized carrier concentrations for n-type and p-type are determined to be 10^19^ and 5 × 10^20^ cm^−3^, respectively. All CdX materials exhibit similar n-type $S^{2}\sigma$ values around 4–5 μW/cmK^2^. The maximum n-type $S^{2}\sigma$ for CdTe (20 μW/cmK^2^) is twice as much as that of CdSe (10 μW/cmK^2^). The p-type $S^{2}\sigma$ value of CdX is comparable to well-known thermoelectric materials such as $Cu_{2}Se$ [40] and SnSe [41], indicating their superior p-type thermoelectric properties. This analysis implies that doping can effectively maintain the electronic transport properties of CdX.

### 3.2. Phononic Properties

Next, we shift our focus to the phonon dispersion in Figure 5. In the low-frequency range, all structures exhibit similar phonon dispersions, as depicted in Figure 5a,c,e,g. All structures show non-negative values in their phonon dispersions, confirming their stability. Upon doping Te atoms into $Cd_{8}Se_{8}$, phonon modes around 130 cm^−1^ moving to a higher frequency. However, for $Cd_{8}Te_{8}$, doping Se atoms leads to phonon modes around 175 cm^−1^ moving to lower frequency. This phenomenon can be seen more clearly in the phonon DOS in Figure 5b,d,f,h. In the phonon DOS, there are two peaks around 130 cm^−1^ in $Cd_{8}Se_{8}$. After doping Te atoms, the peak around 130 cm^−1^ disappears and new peaks at 150 cm^−1^ arise. From the projected phonon DOS, it is evident that the new peaks at 150 cm^−1^ are contributed by Te atoms. The case for $Cd_{8}Te_{8}$ is similar. When doping Se atoms, the peak around 175 cm^−1^ disappears and new peaks contributed by Se atoms at 150 cm^−1^ arise. An increase in the strength of interatomic bonds leads to an increase in the vibration frequency near the atom’s position [42,43].

We further investigate the thermal transport properties of CdX and their corresponding doping systems. Figure 6a shows $\kappa_{l}$ at different temperatures. $\kappa_{l}$ decreases with temperature due to stronger phonon–phonon scattering. $Cd_{8}Se_{8}$ exhibit lower $\kappa_{l}$ values than $Cd_{8}Te_{8}$, as stronger bonds tend to transfer more heat, leading to higher thermal conductivity in crystal structures with stronger bonds. After doping, $\kappa_{l}$ is reduced. Specifically, the $\kappa_{l}$ of $Cd_{8}Se_{7}Te$ is much lower than that of $Cd_{8}Te_{7}Se$, with the former being 0.5 Wm^−1^K^−1^ and the latter being 0.8 Wm^−1^K^−1^ at 300 K. Furthermore, we illustrate the cumulative $\kappa_{l}$ as a function of frequency at 300 K in Figure 6b. For undoped CdX, the rate of increase in $\kappa_{l}$ begins to decrease at 75 cm^−1^, while the node at which the rate of increase slows down after doping drops to 50 cm^−1^. This indicates that doping not only reduces $\kappa_{l}$, but also lowers the frequency at which the maximum rate of $\kappa_{l}$ increase occurs, further contributing to the reduction in $\kappa_{l}$.

To further elucidate the reasons for the behavior of $\kappa_{l}$, we calculate the phonon group velocity ($v_{g}$), phonon lifetime ($\tau_{ph}$), and Grüneisen parameter (γ) as a function of frequency at 300 K as shown in Figure 7. In thermal transport, γ represents the sensitivity of a material’s phonon frequency to changes in volume or pressure, providing insight into the strength of anharmonic scattering. A large value of |γ| indicates the potential for strong phonon–phonon anharmonic scattering [44]. From Figure 7a–d, it can be observed that the doped $v_{g}$ remain largely unchanged, especially in the low-frequency region, consistent with the earlier phonon spectral variations. The speed at which energy is propagated through a material’s lattice vibrations correlates with the distribution of vibrational modes across different frequencies. After doping, a significant decrease in the low-frequency $\tau_{ph}$ is observed in Figure 7e–h. Figure 6b indicates that thermal conductivity is primarily influenced by low-frequency phonons. Subsequently, we analyze the γ in Figure 7i–l, which describes the strength of anharmonic scattering. It is found that the γ also decreases, indicating an enhancement in the strength of anharmonic scattering. This phenomenon evidently arises from the presence of dopant elements.

### 3.3. Mechanical Properties and Bonding Analysis

Figure 8a illustrates the calculated Young’s modulus and shear modulus. It is evident that both Young’s modulus and shear modulus gradually decrease with an increase in the ratio of Te content. These mechanical properties are indicative of the averaged bonding strength within the material. As the Te content increases, the averaged bonding strength of CdX diminishes. Furthermore, we conduct a comprehensive analysis of the bonding in CdX using the COHP method, as depicted in Figure 8b,c. In this context, negative COHP values indicate bonding states, while positive COHP values indicate antibonding states. It is noteworthy that all structures exhibit a similar COHP phenomenon. Upon doping, states for Cd–Se in $Cd_{8}Te_{7}Se$ and Cd–Te in $Cd_{8}Se_{7}Te$ exhibit slightly enhanced strength compared to pristine states. Doping introduces new antibonding states. Below the Fermi level, bonding and antibonding states appear alternately. Notably, bonding states are stronger than antibonding states, thus ensuring the stability of the structure. Additionally, antibonding states typically possess higher energy levels than bonding states, contributing to a reduction in thermal conductivity [45]. This phenomenon arises from the greater delocalization of electrons in antibonding states, which results in reduced heat-carrying capacity compared to bonding states. Consequently, this accounts for the relatively low lattice thermal conductivity observed in CdX.

### 3.4. Thermoelectric Properties

Figure 9 illustrates maximum $S^{2}\sigma$ and ZT values at different temperatures. The $S^{2}\sigma_{max}$ values for doped and pristine CdX are comparable across temperature ranges as shown in Figure 9a,b. As temperature increases, the n-type $S^{2}\sigma_{max}$ values exhibit further enhancement, with $S^{2}\sigma_{max}$ reaching 9.8 μW/cm^−2^ at 900 K for $Cd_{8}Se_{7}Te$. In contrast, there is no clear increasing trend observed for p-type $S^{2}\sigma_{max}$. P-type $S^{2}\sigma_{max}$ values are much higher than n-type ones. Notably, for $Cd_{8}Te_{8}$ and $Cd_{8}Te_{7}Se$, p-type $S^{2}\sigma_{max}$ is approximately 25 μW/cm^−2^, while for $Cd_{8}Se_{8}$ and $Cd_{8}Se_{7}Te$, it is around 15 μW/cm^−2^. Doping does not degrade the $S^{2}\sigma_{max}$ for CdX, as indicated by our findings.

When combined with electronic and phononic transport properties, the $ZT_{max}$ values listed in Figure 9c,d demonstrate high thermoelectric performance for CdX at high temperatures. Despite the lack of increase in n-tpye $S^{2}\sigma_{max}$ with temperature, there is lower lattice thermal conductivity with temperatures resulting in higher $ZT_{max}$ values for doped systems compared to pristine ones. Furthermore, these values can be further enhanced with increasing temperature, with p-type $ZT_{max}$ values reaching up to 3.13 at 900 K. Our results suggest that doped CdX (X=Te, Se) presents potential for realizing both n-type and p-type thermoelectric materials for high-temperature applications.

## 4. Conclusions

In this study, we investigated the electronic, carrier, phonon transport, and thermoelectric properties of isovalent doped CdX (X=Te, Se) compounds using first-principles calculations with the Boltzmann transport equation. Due to the similar properties of Te and Se, the band structures remain nearly unchanged except for the band gaps in doped CdX. The bandgaps are 0.472 eV, 0.372 eV, 0.58 eV, and 0.441 eV for $Cd_{8}Se_{8}$, $Cd_{8}Se_{7}Te$, $Cd_{8}Te_{8}$, and ${Cd}_{8}{Te}_{7}Se$, respectively. Electronic transport properties of CdX are comparable for doped and pristine compounds. However, doping significantly reduces lattice thermal conductivity due to the introduction of impurity scattering. The maximum p-type (n-type) ZT values at 900 K are 1.5 (0.84), 2.51 (1.21), 2.1 (0.88), and 3.13 (1.33) for $Cd_{8}Se_{8}$, $Cd_{8}Se_{7}Te$, $Cd_{8}Te_{8}$, and ${Cd}_{8}{Te}_{7}Se$, respectively. Our study focuses on investigating the impact of isovalent doping in enhancing the thermoelectric properties of materials. Isovalent doping, such as with selenides and tellurides, can maintain electronic transport properties while effectively scattering phonons and decreasing lattice thermal conductivity. Future investigations could explore the potential of decoupling thermoelectric properties through homoelement doping.

## Figures and Tables

**Figure 1 materials-17-01797-f001:**
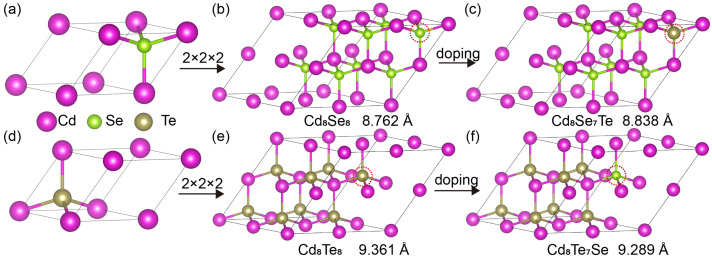
The unit cell of (**a**) CdSe and (**d**) CdTe. The 2×2×2 supercell of (**b**) $Cd_{8}Se_{8}$ and (**e**) $Cd_{8}Te_{8}$. The doped structures of (**c**) $Cd_{8}Se_{7}Te$ and (**f**) $Cd_{8}Te_{7}Se$.

**Figure 2 materials-17-01797-f002:**
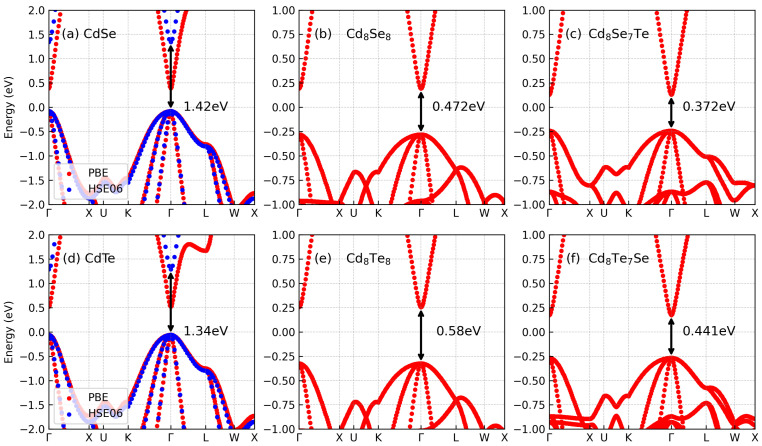
The band structures of (**a**) CdSe, (**b**) $Cd_{8}Se_{8}$, (**c**) $Cd_{8}Se_{7}Te$, (**d**) CdSe, (**e**) $Cd_{8}Se_{8}$, and (**f**) $Cd_{8}Se_{7}Te$.

**Figure 3 materials-17-01797-f003:**
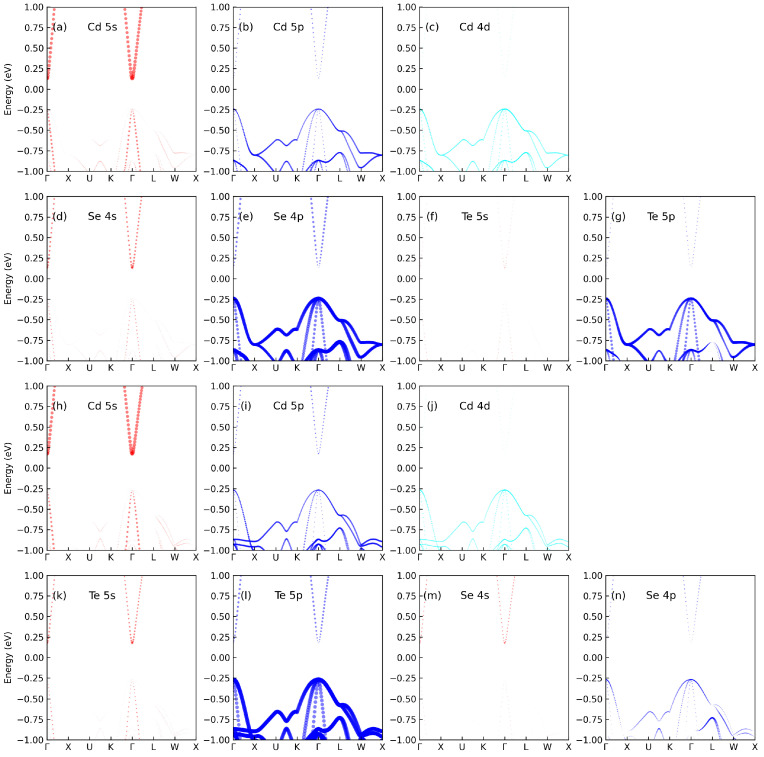
The projected band structure of (**a**) Cd 5s, (**b**) Cd 5p, (**c**) Cd 4d, (**d**) Se 4s, (**e**) Se 4p, (**f**) Te 5s, and (**g**) Te 5p of $Cd_{8}Se_{7}Te$. The projected band structure of (**h**) Cd 5s, (**i**) Cd 5p, (**j**) Cd 4d, (**k**) Te 5s, (**l**) Te 5p, (**m**) Se 4s, and (**n**) Se 4p of $Cd_{8}Te_{7}Se$.

**Figure 4 materials-17-01797-f004:**
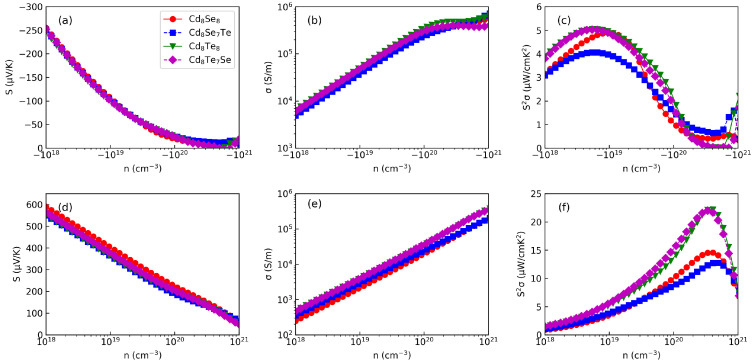
The electronic transport properties of (**a**,**d**) *S*, and (**b**,**e**) σ, and (**c**,**f**) $S^{2}\sigma$.

**Figure 5 materials-17-01797-f005:**
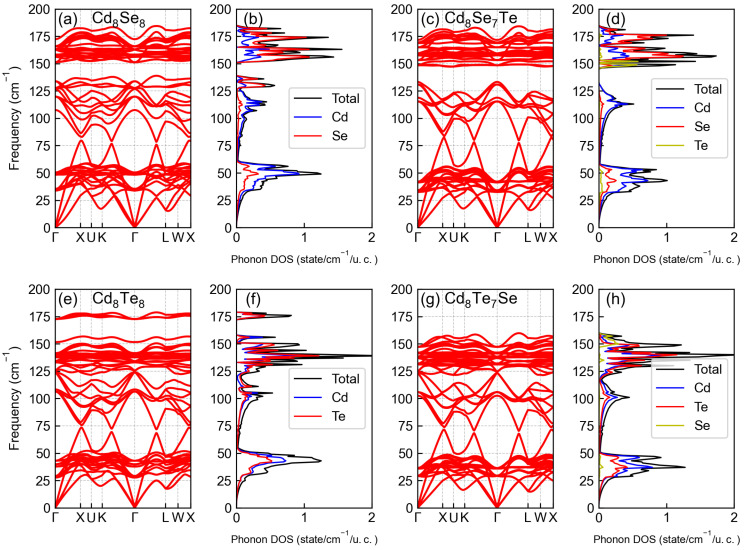
The phonon dispersions for (**a**) $Cd_{8}Se_{8}$, (**c**) $Cd_{8}Se_{7}Te$, (**e**) $Cd_{8}Te_{8}$, and (**g**) $Cd_{8}Te_{7}Se$. The total and projected phononic density of states for (**b**) $Cd_{8}Se_{8}$, (**d**) $Cd_{8}Se_{7}Te$, (**f**) $Cd_{8}Te_{8}$, and (**h**) $Cd_{8}Te_{7}Se$.

**Figure 6 materials-17-01797-f006:**
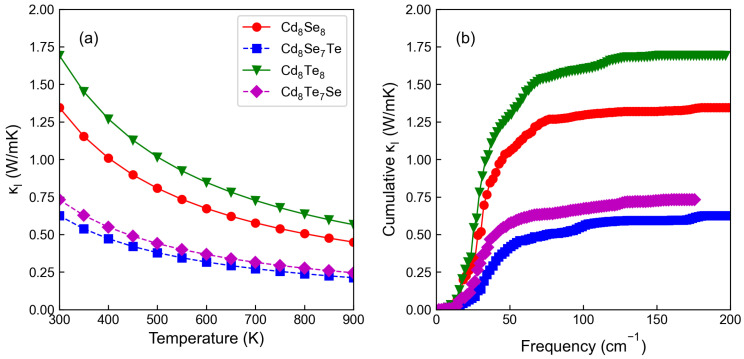
(**a**) The lattice thermal conductivity as a function of temperature. (**b**) Cumulative lattice thermal conductivity as a function of frequency at 300 K.

**Figure 7 materials-17-01797-f007:**
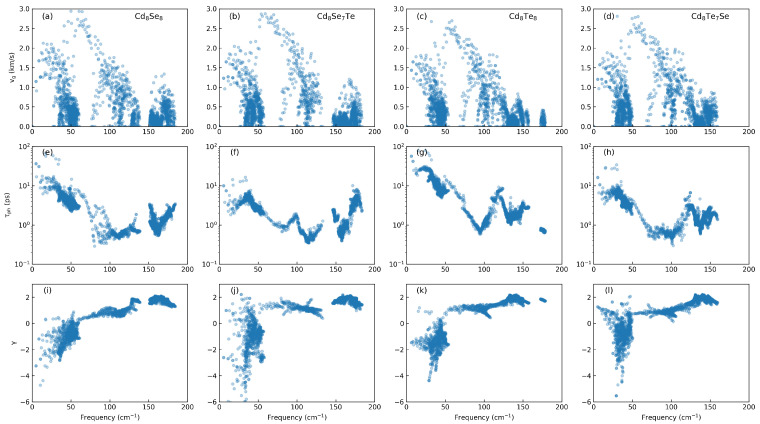
The (**a**–**d**) phonon group velocity, (**e**–**h**) phonon lifetime, and (**i**–**l**) Grüneisen parameter as a function of frequency at 300 K.

**Figure 8 materials-17-01797-f008:**
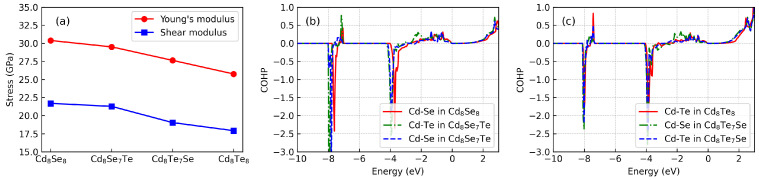
(**a**) Elasctic properties for CdX. COHP for (**b**) $Cd_{8}Se_{7}Te$ and (**c**) $Cd_{8}Te_{7}Se$.

**Figure 9 materials-17-01797-f009:**
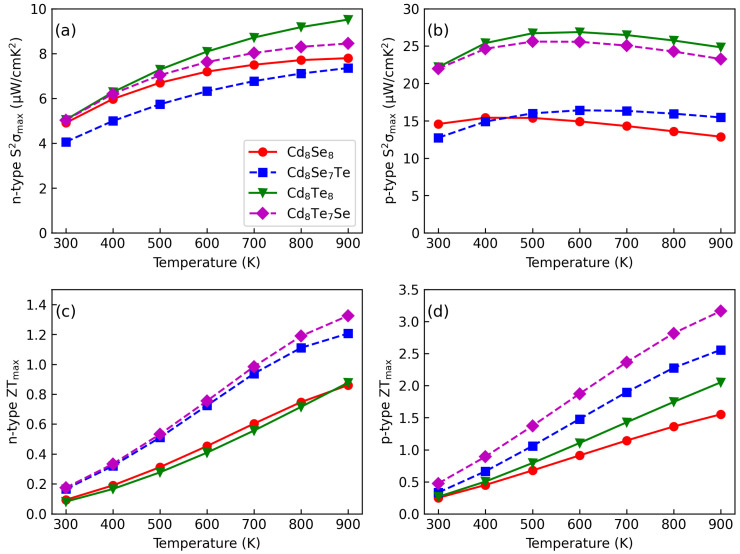
The maximum (**a**) n-type, (**b**) p-type $S^{2}\sigma$ and (**c**) n-type, (**d**) p-type ZT values as a function of temperatures.

## Data Availability

The data provided in this study could be released upon reasonable request.

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
