# Peer review of "First-Principles Study of Doped CdX(X = Te, Se) Compounds: Enhancing Thermoelectric Properties"

_materials, 2024, doi:10.3390/ma17081797_

Round 1

Reviewer 1 Report

Comments and Suggestions for Authors

This paper is on the computational work on doped CdX (X = Te, Ce) compounds for enhancing their thermodynamics properties. The authors are using the well-known periodic density function theory code Quantum Espresso and its extensions to calculate, among others, optimal geometries, electronic information, phonon dispersions, and thermal conductivities. The paper contains extensive calculations, is well-written, and easy for the reader to follow. This work is appropriate for the journal and the paper should be accepted after following my suggestions below (minor revisions).

1)        The authors are using the well-known PBE functional for their DFT calculations. How accurate are their PBE calculated band gaps? Will a hybrid functional such as the HSE06 improve these gaps? The authors need to discuss this. If computational facilities permit, I suggest that an HSE06 calculation is performed in the undoped 2x2x2 system for comparing with the PBE calculated gaps.

2)      It is hard to see the individual graphs in Figure 3. This needs to be revised.

Author Response

We appreciate the reviewer for their valuable feedback. A comprehensive response has been provided in the attached PDF document.

Reviewer 2 Report

Comments and Suggestions for Authors

The presented paper fulfils the Journal Scopus. This article is based on 41 articles in the literature. The authors have undertaken a rather difficult, yet important task from the point of view of materials engineering. Using computational techniques to predict how a material will behave and how its properties will change is quite a challenge. Nevertheless, the authors were up to the task.  The crowning element of the calculations performed was at least the comparison of the obtained data with experimental results (at least one or two properties). This paper contains a fairly comprehensive analysis of structure modification.

Apart from minor editorial errors and a minor linguistic correction, I believe that the manuscript presented may be accepted after minor revision.

Minor corrections need to be made:

  • An important element missing from the article as a whole is the lack of reference to real data (our own experimental or literature reports). How do the theoretical results obtained compare with the values obtained?

  • There is a lack of a summary that states in which direction further research should be carried out. Is the chosen type of admixture the correct option? The use of selenides and telurides in thermoelectric systems is not new.
  • comparison of the data obtained to experimental results (at least one or two properties).

Author Response

(The authors gave the same response as above.)
